# Positional Normalization

**Boyi Li**[1,2]*, **Felix Wu**[1]*, **Kilian Q. Weinberger**[1], **Serge Belongie**[1,2]
[1]Cornell University [2]Cornell Tech
{bl728, fw245, kilian, sjb344}@cornell.edu

## Abstract

A popular method to reduce the training time of deep neural networks is to normalize activations at each layer. Although various normalization schemes have been proposed, they all follow a common theme: normalize across spatial dimensions and discard the extracted statistics. In this paper, we propose an alternative normalization method that noticeably departs from this convention and normalizes exclusively across channels. We argue that the channel dimension is naturally appealing as it allows us to extract the first and second moments of features extracted at a particular image position. These moments capture structural information about the input image and extracted features, which opens a new avenue along which a network can benefit from feature normalization: Instead of disregarding the normalization constants, we propose to re-inject them into later layers to preserve or transfer structural information in generative networks.

## 1 Introduction

A key innovation that enabled the undeniable success of deep learning is the internal normalization of activations. Although normalizing inputs had always been one of the "tricks of the trade" for training neural networks [38], batch normalization (BN) [28] extended this practice to every layer, which turned out to have crucial benefits for deep networks. While the success of normalization methods was initially attributed to "reducing internal covariate shift" in hidden layers [28, 40], an array of recent studies [1, 2, 4, 24, 47, 58, 67, 75] has provided evidence that BN changes the loss surface and prevents divergence even with large step sizes [4], which accelerates training [28].

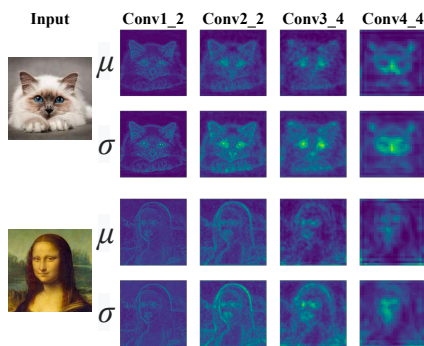

Figure 1: The mean $\mu$ and standard deviation $\sigma$ extracted by PONO at different layers of VGG-19 capture structural information from the input images.

Multiple normalization schemes have been proposed, each with its own set of advantages: Batch normalization [28] benefits training of deep networks primarily in computer vision tasks. Group normalization [72] is often the first choice for small mini-batch settings such as object detection and instance segmentation tasks. Layer Normalization [40] is well suited to sequence models, common in natural language processing. Instance normalization [66] is widely used in image synthesis owing to its apparent ability to remove style information from the inputs. However, all aforementioned normalization schemes follow a common theme: they normalize across spatial dimensions and discard the extracted statistics. The philosophy behind their design is that the first two moments are considered expendable and should be removed.

In this paper, we introduce Positional Normalization (PONO), which normalizes the activations at each position independently across the channels. The extracted mean and standard deviation capture

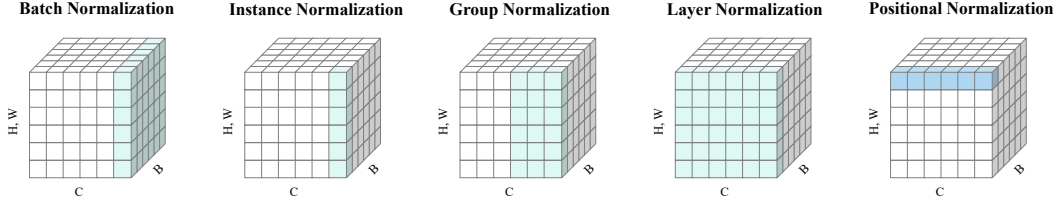

Figure 2: Positional Normalization together with previous normalization methods. In the figure, each subplot shows a feature map tensor, with $B$ as the batch axis, $C$ as the channel axis, and $(H, W)$ as the spatial axis. The entries colored in **green** or **blue** (ours) are normalized by the same mean and standard deviation. Unlike previous methods, our method processes each position independently, and compute both statistics across the channels.

the coarse structural information of an input image (see Figure 1). Although removing the first two moments does benefit training, it also eliminates important information about the image, which — in the case of a generative model — would have to be painfully relearned in the decoder. Instead, we propose to bypass and inject the two moments into a later layer of the network, which we refer to as Moment Shortcut (MS) connection.

PONO is complementary to previously proposed normalization methods (such as BN) and as such can and should be applied jointly. We provide evidence that PONO has the potential to substantially enhance the performance of generative models and can exhibit favorable stability throughout the training procedure in comparison with other methods. PONO is designed to deal with spatial information, primarily targeted at generative [19, 29] and sequential models [23, 32, 56, 63]. We explore the benefits of PONO with MS in several initial experiments across different model architectures and image generation tasks and provide code online at https://github.com/Boyiliee/PONO.

## 2 Related Work

Normalization is generally applied to improve convergence speed during training [50]. Normalization methods for neural networks can be roughly categorized into two regimes: *normalization of weights* [49, 53, 57, 71] and *normalization of activations* [28, 30, 36, 40, 46, 48, 59, 66, 72]. In this work, we focus on the latter.

Given the activations $X \in \mathbb{R}^{B \times C \times H \times W}$ (where $B$ denotes the batch size, $C$ the number of channels, $H$ the height, and $W$ the width) in a given layer of a neural net, the normalization methods differ in the dimensions over which they compute the mean and variance, see Figure 2. In general, activation normalization methods compute the mean $\mu$ and standard deviation (std) $\sigma$ of the features in their own manner, normalize the features with these statistics, and optionally apply an affine transformation with parameters $\beta$ (new mean) and $\gamma$ (new std). This can be written as

$$X'_{b,c,h,w} = \gamma \left( \frac{X_{b,c,h,w} - \mu}{\sigma} \right) + \beta. \tag{1}$$

Batch Normalization (BN) [28] computes $\mu$ and $\sigma$ across the B, H, and W dimensions. BN increases the robustness of the network with respect to high learning rates and weight initializations [4], which in turn drastically improves the convergence rate. Synchronized Batch Normalization treats features of mini-batches across multiple GPUs like a single mini-batch. Instance Normalization (IN) [66] treats each instance in a mini-batch independently and computes the statistics across only spatial dimensions (H and W). IN aims to make a small change in the stylization architecture results in a significant qualitative improvement in the generated images. Layer Normalization (LN) normalizes all features of an instance within a layer jointly, i.e., calculating the statistics over the C, H, and W dimensions. LN is beneficial in natural language processing applications [40, 68]. Notably, none of the aforementioned methods normalize the information at different spatial position independently. This limitation gives rise to our proposed Positional Normalization.

Batch Normalization introduces two learned parameters $\beta$ and $\gamma$ to allow the model to adjust the mean and std of the post-normalized features. Specifically, $\beta, \gamma \in \mathbb{R}^C$ are channel-wise parameters. Conditional instance normalization (CIN) [15] keeps a set parameter of pairs $\{(\beta_i, \gamma_i) | i \in \{1, \ldots, N\}\}$

which enables the model to have $N$ different behaviors conditioned on a style class label $i$. Adaptive instance normalization (AdaIN) [26] generalizes this to an infinite number of styles by using the $\mu$ and $\sigma$ of IN borrowed from another image as the $\beta$ and $\gamma$. Dynamic Layer Normalization (DLN) [35] relies on a neural network to generate the $\beta$ and $\gamma$. Later works [27, 33] refine AdaIN and generate the $\beta$ and $\gamma$ of AdaIN dynamically using a dedicated neural network. Conditional batch normalization (CBN) [10] follows a similar spirit and uses a neural network that takes text as input to predict the residual of $\beta$ and $\gamma$, which is shown to be beneficial to visual question answering models.

Notably, all aforementioned methods generate $\beta$ and $\gamma$ as vectors, shared across spatial positions. In contrast, Spatially Adaptive Denormalization (SPADE) [52], an extension of Synchronized Batch Normalization with dynamically predicted weights, generates the spatially dependent $\beta, \gamma \in \mathbb{R}^{B \times C \times H \times W}$ using a two-layer ConvNet with raw images as inputs.

Finally, we introduce shortcut connections to transfer the first and second moment from early to later layers. Similar skip connections (with add, concat operations) have been introduced in ResNets [20] and DenseNets [25] and earlier works [3, 23, 34, 54, 62], and are highly effective at improving network optimization and convergence properties [43].

# 3 Positional Normalization and Moment Shortcut

Prior work has shown that feature normalization has a strong beneficial effect on the convergence behavior of neural networks [4]. Although we agree with these findings, in this paper we claim that removing the first and second order information at multiple stages throughout the network may also deprive the deep net of potentially useful information — particularly in the context of generative models, where a plausible image needs to be generated.

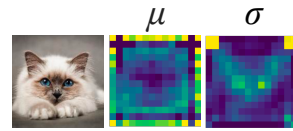

Figure 3: PONO statistics of DenseBlock-3 of a pretrained DenseNet-161.

**PONO.** Our normalization scheme, which we refer to as *Positional Normalization (PONO)*, differs from prior work in that we normalize exclusively over the channels at any given fixed pixel location (see Figure 2). Consequently, the extracted statistics are position dependent and reveal structural information at this particular layer of the deep net. The mean $\mu$ can be considered itself an "image", where the intensity of pixel $i, j$ represents the average activation at this particular image location in this layer. The standard deviation $\sigma$ is the natural second order extension. Formally, PONO computes

$$\mu_{b,h,w} = \frac{1}{C} \sum_{c=1}^{C} X_{b,c,h,w}, \quad \sigma_{b,h,w} = \sqrt{\frac{1}{C} \sum_{c=1}^{C} \left( X_{b,c,h,w} - \mu_{b,h,w} \right)^2 + \epsilon}, \tag{2}$$

where $\epsilon$ is a small stability constant (*e.g.*, $\epsilon = 10^{-5}$) to avoid divisions by zero and imaginary values due to numerical inaccuracies.

**Properties.** As PONO computes the normalization statistics at all spatial positions independently from each other (unlike BN, LN, CN, and GN) it is translation, scaling, and rotation invariant. Further, it is complementary to existing normalization methods and, as such, can be readily applied in combination with e.g. BN.

**Visualization.** As the extracted mean and standard deviations are themselves images, we can visualize them to obtain information about the extract features at the various layers of a convolutional network. Such visualizations can be revealing and could potentially be used to debug or improve network architectures. Figure 1 shows heat-maps of the $\mu$ and $\sigma$ captured by PONO at several layers (Conv1_2, Conv2_2, Conv3_4, and Conv4_4) of VGG-19 [60]. The figure reveals that the features in lower layers capture the silhouette of a cat while higher layers locate the position of noses, eyes, and the end points of ears —- suggesting that later layers may focus on higher level concepts corresponding to essential facial features (eyes, nose, mouth), whereas earlier layers predominantly extract generic low level features like edges. We also observe a similar phenomenon from the features of ResNets [20] and DenseNets [25] (see Figure 3 and Appendix). The resulting images are reminiscent of related statistics captured in texture synthesis [14, 16–18, 21, 51, 70]. We observe that unlike VGG and ResNet, DenseNet exhibits strange behavior on corners and boundaries which may

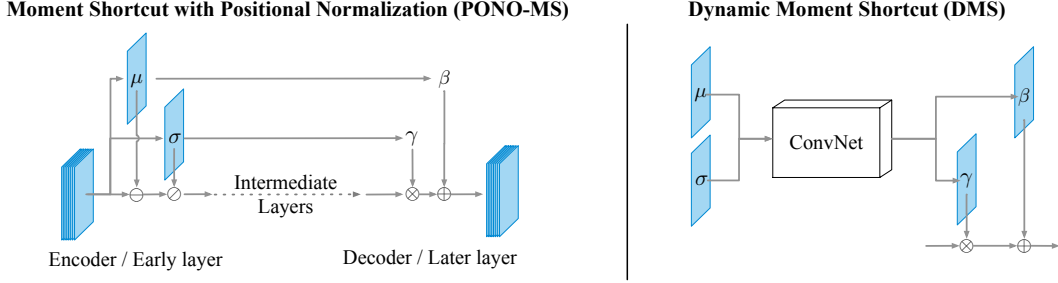

**Moment Shortcut with Positional Normalization (PONO-MS)**

$\mu$

$\beta$

$\sigma$

$\gamma$

Intermediate
Layers

Encoder / Early layer

Decoder / Later layer

**Dynamic Moment Shortcut (DMS)**

$\mu$

$\beta$

ConvNet

$\sigma$

$\gamma$

Figure 4: Left: PONO-MS directly uses the extracted mean and standard deviation as $\beta$ and $\gamma$. Right: Optionally, one may use a (shallow) ConvNet to predict $\beta$ and $\gamma$ dynamically based on $\mu$ and $\sigma$.

degrade performance when fine-tuned on tasks requiring spatial information such as object detection or segmentation. This suggests that the padding and downsampling procedure of DenseNet should be revisited and may lead to improvements if fixed, see Figure 3. The visualizations of the PONO statistics support our hypothesis that the mean $\mu$ and the standard deviation $\sigma$ may indeed capture structural information of the image and extracted features, similar to the way statistics computed by IN have the tendency to capture aspects of the style of the input image [26, 66]. This extraction of valuable information motivates the Moment Shortcut described in the subsequent section.

### 3.1 Moment Shortcut

In generative models, a deep net is trained to generate an output image from some inputs (images). Typically, generative models follow an encoder-decoder architecture, where the encoder digests an image into a condensed form and the decoder recovers a plausible image with some desired properties. For example, Huang et al. [26] try to transfer the style from an image A to an image B, Zhu et al. [77] "translate" an image from an input distribution (e.g., images of zebras) to an output distribution (e.g., images of horses), Choi et al. [8] use a shared encoder-decoder with a classification loss in the encoded latent space to enable translation across multiple distributions, and [27, 39] combine the structural information of an image with the attributes from another image to generate a fused output.

U-Nets [55] famously achieve strong results and compelling optimization properties in generative models through the introduction of skip connections from the encoder to the decoder. PONO gives rise to an interesting variant of such skip connections. Instead of connecting all channels, we only "fast-forward" the positional moment information $\mu$ and $\sigma$ extracted from earlier layers. We refer to this approach as Moment Shortcut (MS).

**Autoencoders.** Figure 4 (left) illustrates the use of MS in the context of an autoencoder. Here, we extract the first two moments of the activations $(\mu, \sigma)$ in an encoder layer, and send them to a corresponding decoder layer. Importantly, the mean is *added* in the encoder, and the std is *multiplied*, similar to $(\beta, \gamma)$ in the standard BN layer. To be specific, $\mathrm{MS}(\mathbf{x}) = \gamma F(\mathbf{x}) + \beta$, where $F$ is modeled by the intermediate layers, and the $\beta$ and $\gamma$ are the $\mu$ and $\sigma$ extracted from the input $\mathbf{x}$. MS biases the decoder explicitly so that the activations in the decoder layers give rise to similar statistics than corresponding layers in the encoder. As MS shortcut connections can be used with and without normalization, we refer to the combination of PONO with MS as **PONO-MS** throughout.

Provided PONO does capture essential structural signatures from the input images, we can use the extracted moments to transfer this information from a source to a target image. This opens an opportunity to go beyond autoencoders and use PONO-MS in image-to-image translation settings, for example in the context of CycleGAN [77] and Pix2Pix [29]. Here, we transfer the structure (through $\mu$ and $\sigma$) of one image from the encoder to the decoder of another image.

**Dynamic Moment Shortcut.** Inspired by Dynamic Layer Normalization and similar works [6, 27, 33, 35, 52], we propose a natural extension called Dynamic Moment Shortcut (DMS): instead of re-injecting $\mu$ and $\sigma$ as is, we use a convolutional neural network that takes $\mu$ and $\sigma$ as inputs to generate the $\beta$ and $\gamma$ for MS. This network can either generate one-channel outputs $\beta, \gamma \in \mathcal{R}^{B \times 1 \times H \times W}$ or multi-channel outputs $\beta, \gamma \in \mathcal{R}^{B \times C \times H \times W}$ (like [52]). The right part of Figure 4 illustrates DMS with one-channel output. DMS is particularly helpful when the task involves shape deformation or

distortion. We refer to this approach as **PONO-DMS** in the following sections. In our experiments, we explore using a ConvNet with either one or two layers.

# 4 Experiments and Analysis

We conduct our experiments on unpaired and paired image translation tasks using CycleGAN [77] and Pix2pix [29] as baselines, respectively. Our code is available at https://github.com/Boyiliee/PONO.

## 4.1 Experimental Setup

We follow the same setup as CycleGAN [77] and Pix2pix [29] using their official code base.[2] We use four datasets: 1) **Maps** (Maps $\leftrightarrow$ aerial photograph) including 1096 training images scraped from Google Maps and 1098 images in each domain for testing. 2) **Horse $\leftrightarrow$ Zebra** including 1067 horse images and 1334 zebra images downloaded from ImageNet [11] using keywords wild horse and zebra, and 120 horse images and 140 zebra images for testing. 3) **Cityscapes** (Semantic labels $\leftrightarrow$ photos) [9] including 2975 images from the Cityscapes training set for training and 500 images in each domain for testing. 4) **Day $\leftrightarrow$ Night** including 17,823 natural scene images from Transient Attributes dataset [37] for training, and 2,287 images for testing. The first, third, and fourth are paired image datasets; the second is an unpaired image dataset. We use the first and second for CycleGAN, and all the paired-image datasets for Pix2pix.

**Evaluation metrics.** We use two evaluation metrics, as follows. (1) Fréchet Inception Distance [22] between the output images and all test images in the target domain. FID uses an Inception [64] model pretrained on ImageNet [11] to extract image features. Based on the means and covariance matrices of the two sets of extracted features, FID is able to estimate how different two distributions are. (2) Average Learned Perceptual Image Patch Similarity distance [76] of all output and target image pairs. LPIPS is based on pretrained AlexNet [36] features[3], which has been shown [76] to be highly correlated to human judgment.

**Baselines.** We include four baseline approaches: (1) CycleGAN or Pix2pix baselines; (2) these baselines with SPADE [52], which passes the input image through a 2-layer ConvNet and generates the $\beta$ and $\gamma$ for BN in the decoder. (3) the baseline with *additive skip connections* where encoder activations are added to decoder activations; (4) the baseline with *concatenated skip connections*, where encoder activations are concatenated to decoder activations as additional channels (similar to U-Nets [55]). For all models, we follow the same setup as CycleGAN [77] and Pix2pix [29] using their implementations. Throughout we use the hyper-parameters suggested by the original authors.

## 4.2 Comparison against Baselines

We add PONO-MS and PONO-DMS to the CycleGAN generator; see the Appendix for the model architecture. Table 1 shows that both cases outperform all baselines at transforming maps into photos, with the only exception of SPADE (which however performs worse in the other direction).

Although skip connections could help make up for the lost information, we postulate that directly adding the intermediate features back may introduce too much unnecessary information and might distract the model. Unlike the skip connections, SPADE uses the input to predict the parameters for normalization. However, on Photo $\rightarrow$ Map, the model has to learn to compress the input photos and extract structural information from it. A re-introduction of the original raw input may disturb this process and explain the worse performance. In contrast, PONO-MS normalizes exclusively across channels which allows us to capture structural information of a particular input image and re-inject/transfer it to later layers.

The Pix2pix model [29] is a conditional adversarial network introduced as a general-purpose solution for image-to-image translation problems. Here we conduct experiments on whether PONO-MS helps Pix2pix [29] with Maps [77], Cityscapes [9] and Day $\leftrightarrow$ Night [37]. We train for 200 epochs and compare the results with/without PONO-MS, under similar conditions with matching number of parameters. Results are summarized in Table 2.

| Method | # of param. | Map → Photo FID | Photo → Map FID | Horse → Zebra FID | Zebra → Horse FID |
|---|---|---|---|---|---|
| CycleGAN (Baseline) | 2×11.378M | 57.9 | 58.3 | 86.3 | 155.9 |
| +Skip Connections | **+0M** | 83.7 | 56.0 | 75.9 | 145.5 |
| +Concatenation | +0.74M | 58.9 | 61.2 | 85.0 | 145.9 |
| +SPADE | +0.456M | **48.2** | 59.8 | 71.2 | 159.9 |
| +PONO-MS | **+0M** | 52.8 | **53.2** | 71.2 | 142.2 |
| +PONO-DMS | +0.018M | 53.7 | 54.1 | **65.7** | **140.6** |

Table 1: FID of CycleGAN and its variants on Map ↔ Photo and Zebra ↔ Horse datasets. CycleGAN is trained with two directions together, it is essential to have good performance in both directions.

| | Maps [77] | | Cityscapes [9] | | Day ↔ Night [37] | |
|---|---|---|---|---|---|---|
| | Map → Photo | Photo → Map | SL → Photo | Photo → SL | Day → Night | Night →Day |
| Pix2pix (Baseline) | 60.07 / **0.333** | 68.73 / 0.169 | 71.24 / 0.422 | 102.38 / **0.223** | 196.58 / 0.608 | 131.94 / **0.531** |
| +PONO-MS | **56.88** / **0.333** | **68.57** / **0.166** | **60.40** / **0.331** | **97.78** / 0.224 | **191.10** / **0.588** | **131.83** / 0.534 |

Table 2: Comparison based on Pix2pix by FID / LPIPS on Maps [77], Cityscapes [9], and Day2Night. Note: for all scores, the lower the better (SL is short for *Semantic labels*).

## 4.3 Ablation Study

Table 3 contains the results of several experiments to evaluate the sensitivities and design choices of PONO-MS and PONO-DMS. Further, we evaluate *Moment Shortcut (MS)* without PONO, where we bypass both statistics, $\mu$ and $\sigma$, without normalizing the features. The results indicate that PONO-MS outperforms MS alone, which suggests that normalizing activations with PONO is beneficial. PONO-DMS can lead to further improvements, and some settings (e.g. *1 conv 3 × 3, multi-channel*) consistently outperform PONO-MS. Here, multi-channel predictions are clearly superior over single-channel predictions but we do not observe consistent improvements from a $5 \times 5$ rather than a $3 \times 3$ kernel size.

**Normalizations.** Unlike previous normalization methods such as BN and GN that emphasize on accelerating and stabilizing the training of networks, PONO is used to split off part of the spatial information and re-inject it later. Therefore, PONO-MS can be applied jointly with other normalization methods. In Table 4 we evaluate four normalization approaches (BN, IN, LN, GN) with and without PONO-MS, and PONO-MS without any additional normalization (bottom row). In detail, *BN + PONO-MS* is simply applying *PONO-MS* to the baseline model and keep the original BN modules which have a different purpose: to stabilize and speed up the training. We also show the models where BN is replaced by LN/IN/GN as well as these models with PONO-MS. The last row shows PONO-MS can work independently when we remove the original BN in the model. Each table entry displays the FID score without and with PONO-MS (the lower score is in **bold**). The final column (very right) contains the average improvement across all four tasks, relative to the default architecture, BN without PONO-MS. Two clear trends emerge: 1. All four normalization methods improve with PONO-MS on average and on almost all individual tasks; 2. additional normalization is clearly beneficial over pure PONO-MS (bottom row).

| Method | Map → Photo | Photo → Map | Horse → Zebra | Zebra → Horse |
|---|---|---|---|---|
| CycleGAN (Baseline) | 57.9 | 58.3 | 86.3 | 155.9 |
| +Moment Shortcut (MS) | 54.5 | 56.6 | 79.8 | 146.1 |
| +PONO-MS | 52.8 | 53.2 | 71.2 | 142.2 |
| +PONO-DMS (1 conv $3 \times 3$, one-channel) | 55.1 | 53.8 | 74.1 | 147.2 |
| +PONO-DMS (2 conv $3 \times 3$, one-channel) | 56.0 | 53.3 | 81.6 | 144.8 |
| +PONO-DMS (1 conv $3 \times 3$, multi-channel) | 53.7 | 54.1 | 65.7 | **140.6** |
| +PONO-DMS (2 conv $5 \times 5$, multi-channel) | 52.7 | 54.7 | **64.9** | 155.2 |
| +PONO-DMS (2 conv $3 \times 3, 5 \times 5$, multi-channel) | **48.9** | 57.3 | 74.3 | 148.4 |
| +PONO-DMS (2 conv $3 \times 3$, multi-channel) | 50.3 | **51.4** | 72.2 | 146.1 |

Table 3: Comparisons of ablation study on FID (lower is better). PONO-MS outperforms MS alone. PONO-DMS can help obtain better performance than PONO-MS.

| Method | Map → Photo | Photo → Map | Horse → Zebra | Zebra → Horse | Avg. Improvement |
|---|---|---|---|---|---|
| BN (Default) / BN + PONO-MS | 57.92 / **52.81** | 58.32 / **53.23** | 86.28 / **71.18** | 155.91 / **142.21** | 1 / **0.890** |
| IN / IN + PONO-MS | 67.87 / **47.14** | 57.93 / **54.18** | **67.85** / 69.21 | 154.15 / **153.61** | 0.985 / **0.883** |
| LN / LN + PONO-MS | 54.84 / **49.81** | 53.00 / **50.08** | 87.26 / **67.63** | 154.49 / **142.05** | 0.964 / **0.853** |
| GN / GN + PONO-MS | 51.31 / **50.12** | 50.62 / **50.50** | 93.58 / **63.53** | **143.56** / 144.99 | 0.940 / **0.849** |
| PONO-MS | **49.59** | **52.21** | **84.68** | **143.47** | **0.913** |

Table 4: FID scores (lower is better) of CycleGAN with different normalization methods.

# 5  Further Analysis and Explorations.

In this section, we apply PONO-MS to two state-of-the-art unsupervised image-to-image translation models: MUNIT [27] and DRIT [39]. Both approaches may arguably be considered concurrent works and share a similar design philosophy. Both aim to translate an image from a source to a target domain, while imposing the attributes (or the style) of another target domain image.

As task, we are provided with an image $x_A$ in source domain A and an image $x_B$ in target domain B. DRIT uses two encoders, one to extract content features $c_A$ from $x_A$, and the other to extract attribute features $a_B$ from $x_B$. A decoder then takes $c_A$ and $a_B$ as inputs to generate the output image $x_{A \to B}$. MUNIT follows a similar pipeline. Both approaches are trained on the two directions, $A \to B$ and $B \to A$, simultaneously. We apply PONO to DRIT or MUNIT immediately after the first three convolution layers (convolution layers before the residual blocks) of the content encoders. We then use MS before the last three transposed convolution layers with matching decoder sizes. We follow the DRIT and MUNIT frameworks and consider the extracted statistics ($\mu$'s and $\sigma$'s) as part of the content tensors.

## 5.1  Experimental Setup

We consider two datasets provided by the authors of DRIT: 1) **Portrait ↔ Photo** [39, 44] with 1714 painting images and 6352 human photos for training, and 100 images in each domain for testing and 2) **Cat ↔ Dog** [39] containing 771 cat images and 1264 dog images for training, and 100 images in each domain for testing.

In the following experiments, we use the official codebases[4], closely follow their proposed hyper-parameters and train all models for 200K iterations. We use the holdout test images as the inputs for evaluation. For each image in the source domain, we randomly sample 20 images in the target domain to extract the attributes and generate 20 output images. We consider four evaluation metrics: 1) FID [22]: Fréchet Inception Distance between the output images and all test images in the target domain, 2) LPIPS$_{attr}$ [76]: average LPIPS distance between each output image and its corresponding input image in the target domain, 3) LPIPS$_{cont}$: average LPIPS distance between each output image and its input in the source domain, and 4) perceptual loss (VGG) [31, 60]: L1 distance between the VGG-19 Conv4_4 features [7] of each output image and its corresponding input in the source domain. The FID and LPIPS$_{attr}$ are used to estimate how likely the outputs are to belong to the target domain, while LPIPS$_{cont}$ and VGG loss are adopted to estimate how much the outputs preserve the structural information in the inputs. All of them are distance metrics where lower is better. The original implementations of DRIT and MUNIT assume differently sized input images (216x216 and 256x256, respectively), which precludes a direct comparison across approaches.

## 5.2  Results of Attribute Controlled Image Translation

Figure 5 shows the qualitative results on the Cat ↔ Dog dataset. (Here we show the results of MUNIT' + PONO-MS which will be explained later.) We observe a clear trend that PONO-MS helps these two models obtain more plausible results. We observe the models with PONO-MS is able to capture the content features and attributes distributions, which motivates baseline models to digest different information from both domains. For example, in the first row, when translating cat to dog, DRIT with PONO-MS is able to capture the cat's facial expression, and MUNIT with PONO-MS could successfully generate dog images with plausible content, which largely boosts the performance of the baseline models. More qualitative results of randomly selected inputs are provided in the Appendix.

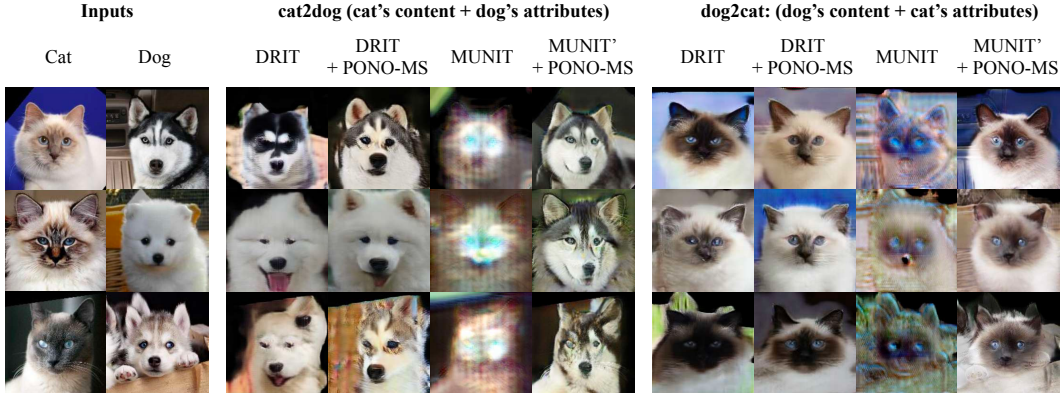

Figure 5: PONO-MS improves the quality of both DRIT [39] and MUNIT [27] on Cat $\leftrightarrow$ Dog.

| | Portrait $\rightarrow$ Photo | | | | Portrait $\leftarrow$ Photo | | | |
| | FID | LPIPS$_{attr}$ | LPIPS$_{cont}$ | VGG | FID | LPIPS$_{attr}$ | LPIPS$_{cont}$ | VGG |
|---|---|---|---|---|---|---|---|---|
| DRIT | 131.2 | 0.545 | 0.470 | 1.796 | 104.5 | 0.585 | 0.476 | 2.033 |
| DRIT + PONO-MS | **127.9** | **0.534** | **0.457** | **1.744** | **99.5** | **0.575** | **0.463** | **2.022** |
| MUNIT | 220.1 | 0.605 | 0.578 | 1.888 | 149.6 | 0.619 | 0.670 | 2.599 |
| MUNIT + PONO-MS | 270.5 | 0.541 | 0.423 | 1.559 | 127.5 | 0.586 | 0.477 | 2.202 |
| MUNIT' | 245.0 | 0.538 | 0.455 | 1.662 | 158.1 | 0.601 | 0.620 | 2.434 |
| MUNIT' + PONO-MS | **159.4** | **0.424** | **0.319** | **1.324** | **125.1** | **0.566** | **0.312** | **1.824** |
| | Cat $\rightarrow$ Dog | | | | Cat $\leftarrow$ Dog | | | |
| | FID | LPIPS$_{attr}$ | LPIPS$_{cont}$ | VGG | FID | LPIPS$_{attr}$ | LPIPS$_{cont}$ | VGG |
| DRIT | **45.8** | 0.542 | 0.581 | **2.147** | 42.0 | 0.524 | **0.576** | 2.026 |
| DRIT + PONO-MS | 47.5 | **0.524** | **0.576** | **2.147** | **41.0** | **0.514** | 0.604 | **2.003** |
| MUNIT | 315.6 | 0.686 | 0.674 | 1.952 | 290.3 | 0.629 | 0.591 | 2.110 |
| MUNIT + PONO-MS | 254.8 | 0.632 | 0.501 | 1.614 | 276.2 | 0.624 | 0.585 | 2.119 |
| MUNIT' | 361.5 | 0.699 | 0.607 | 1.867 | 289.0 | 0.767 | 0.789 | 2.228 |
| MUNIT' + PONO-MS | **80.4** | **0.615** | **0.406** | **1.610** | **90.8** | **0.477** | **0.428** | **1.689** |

Table 5: PONO-MS can improve the performance of MUNIT [27], while for DRIT [39] the improvement is marginal. MUNIT' is MUNIT with one more Conv3x3-LN-ReLU layer before the output layer in the decoder, which introduces $0.2\%$ parameters into the generator. Note: for all scores, the lower the better.

Table 5 show the quantitative results on both Cat $\leftrightarrow$ Dog and Portrait $\leftrightarrow$ Photo datasets. PONO-MS improves the performance of both models on all instance-level metrics (LPIPS$_{attr}$, LPIPS$_{cont}$, and VGG loss). However, the dataset-level metric, FID, doesn't improve too much. We believe the reason is that FID is calculated based on the first two order statistic of Inception features and may discard some subtle differences between each output pair.

Interestingly MUNIT, while being larger than DRIT (30M parameters vs. 10M parameters), doesn't perform better on these two datasets. One reason for its relatively poor performance could be that the model was not designed for these datasets (MUNIT uses a much larger unpublished *dogs to big cats* dataset), the dataset are very small, and the default image resolution is slightly different. To further improve MUNIT + PONO-MS, we add one more Conv3x3-LN-ReLU layer before the output layer. Without this, there is only one layer between the outputs and the last re-introduced $\mu$ and $\sigma$. Therefore, adding one additional layer allows the model to learn a nonlinear function of these $\mu$ and $\sigma$. We call this model MUNIT' + PONO-MS. Adding this additional layer significantly enhances the performance of MUNIT while introducing only 75K parameters (about $0.2\%$). We also provide the numbers of MUNIT' (MUNIT with one additional layer) as a baseline for a fair comparison.

Admittedly, the state-of-the-art generative models employ complex architecture and a variety of loss functions; therefore, unveiling the full potential of PONO-MS on these models can be nontrivial and required further explorations. It is fair to admit that the results of all model variations are still largely unsatisfactory and the image translation task remains an open research problem.

However, we hope that our experiments on DRIT and MUNIT may shed some light on the potential value of PONO-MS, which could open new interesting directions of research for neural architecture design.

## 6   Conclusion and Future Work

In this paper, we propose a novel normalization technique, Positional Normalization (PONO), in combination with a purposely limited variant of shortcut connections, Moment Shortcut (MS). When applied to various generative models, we observe that the resulting model is able to preserve structural aspects of the input, improving the plausibility performance according to established metrics. PONO and MS can be implemented in a few lines of code (see Appendix). Similar to Instance Normalization, which has been observed to capture the style of image [26, 33, 66], Positional Normalization captures structural information. As future work we plan to further explore such disentangling of structural and style information in the design of modern neural architectures.

It is possible that PONO and MS can be applied to a variety of tasks such as image segmentation [45, 55], denoising [41, 73], inpainting [74], super-resolution [13], and structured output prediction [61]. Further, beyond single image data, PONO and MS may also be applied to video data [42, 69], 3D voxel grids [5, 65], or tasks in natural language processing [12].

### Acknowledgments

This research is supported in part by the grants from Facebook, the National Science Foundation (III-1618134, III-1526012, IIS1149882, IIS-1724282, and TRIPODS-1740822), the Office of Naval Research DOD (N00014-17-1-2175), Bill and Melinda Gates Foundation. We are thankful for generous support by Zillow and SAP America Inc.

## Footnotes

*: Equal contribution.

[2]https://github.com/junyanz/pytorch-CycleGAN-and-pix2pix

[3]https://github.com/richzhang/PerceptualSimilarity, version 0.1.

[4]https://github.com/NVlabs/MUNIT/ and https://github.com/HsinYingLee/DRIT

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
