[Supplementary Material · Positional_Normalization_Camera_Ready_supp.pdf]

# Positional Normalization (Supplementary Material)

**Boyi Li**[1,2]*, **Felix Wu**[1]*, **Kilian Q. Weinberger**[1], **Serge Belongie**[1,2]
[1]Cornell University [2]Cornell Tech
{bl728, fw245, kilian, sjb344}@cornell.edu

# Appendices

## A    Algorithm of PONO-MS

The implementation of PONO-MS in TensorFlow [1] an PyTorch[7] are shown in Listing 1 and 2 respectively.

```
# x is the features of shape [B, H, W, C]

# In the Encoder
def PONO(x, epsilon=1e-5):
    mean, var = tf.nn.moments(x, [3], keep_dims=True)
    std = tf.sqrt(var + epsilon)
    output = (x - mean) / std
    return output, mean, std

# In the Decoder
# one can call MS(x, mean, std)
# with the mean and std are from a PONO in the encoder
def MS(x, beta, gamma):
    return x * gamma + beta
```

Listing 1: PONO and MS in TensorFlow

```
# x is the features of shape [B, C, H, W]

# In the Encoder
def PONO(x, epsilon=1e-5):
    mean = x.mean(dim=1, keepdim=True)
    std = x.var(dim=1, keepdim=True).add(epsilon).sqrt()
    output = (x - mean) / std
    return output, mean, std

# In the Decoder
# one can call MS(x, mean, std)
# with the mean and std are from a PONO in the encoder
def MS(x, beta, gamma):
    return x * gamma + beta
```

Listing 2: PONO and MS in PyTorch

## B  Equations of Existing Normalization

Batch Normalization (BN) computes the mean and std across B, H, and H dimensions, i.e.

$$\mu_c = \mathbb{E}_{b,h,w}[X_{b,c,h,w}], \quad \sigma_c = \sqrt{\mathbb{E}_{b,h,w}[X_{b,c,h,w}^2 - \mu_c] + \epsilon},$$

where $\epsilon$ is a small constant applied to handle numerical issues.

Synchronized Batch Normalization views features of mini-batches across multiple GPUs as a single mini-batch.

Instance Normalization (IN) treats each instance in a mini-batch independently and computes the statistics across only spatial dimensions, i.e.

$$\mu_{b,c} = \mathbb{E}_{h,w}[X_{b,c,h,w}], \quad \sigma_{b,c} = \sqrt{\mathbb{E}_{h,w}[X_{b,c,h,w}^2 - \mu_{b,c}] + \epsilon}.$$

Layer Normalization (LN) normalizes all features of an instance within a layer jointly, i.e.

$$\mu_b = \mathbb{E}_{c,h,w}[X_{b,c,h,w}], \quad \sigma_b = \sqrt{\mathbb{E}_{c,h,w}[X_{b,c,h,w}^2 - \mu_b] + \epsilon}.$$

Finally, Group Normalization (GN) lies between IN and LN, it devides the channels into $G$ groups and apply layer normalization within a group. When $G = 1$, GN becomes LN. Conversely, when the $G = C$, it is identical to IN. To define it formally, it computes

$$\mu_{b,g} = \mathbb{E}_{c \in S_g, h, w}[X_{b,c,h,w}], \quad \sigma_{b,g} = \sqrt{\mathbb{E}_{c \in S_g, h, w}[X_{b,c,h,w}^2 - \mu_{b,g}] + \epsilon},$$

where $S_g = \{\lceil \frac{(g-1)C}{G} + 1 \rceil, \ldots, \lceil \frac{gC}{G} \rceil\}$.

## C  PONO Statistics of Models Pretrained on ImageNet

Figure 1 shows the means and the standard deviations extracted by PONO based on the features generated by VGG-19 [8], ResNet-152 [3], and DenseNet-161 [4] pretrained on ImageNet [2].

## D  Implementation details

We add PONO to the encoder right after a convolution operation and before other normalization or nonlinear activation function. Figure 2 shows the model architecture of CycleGAN [9] with Positional Normalization. Pix2pix [5] uses the same architecture.

## E  Qualitative Results Based on CycleGAN and Pix2pix

We show some outputs of CycleGAN in Figure 3. The Pix2pix outputs are shown in Figure 4.

## F  Qualitative Results Based on DRIT and MUNIT.

We randomly sample 10 *cat and dog* image pairs and show the outputs of DRIT, DRIT + PONO-MS, MUNIT, and MUNIT' PONO-MS in Figure 5.

## G  PONO in Image Classification

To evaluate PONO on image classification task, we add PONO to the begining of each ResBlock of ResNet-18 [3] (also affects the shortcut). We followed the common training procedure base on Wei Yang's open sourced code [2] on ImageNet [6]. Figure 6 shows that with PONO, the training loss and error are reduced significantly and the validation error also drops slightly from 30.09 to 30.01. Admittedly, this is not a significant improvement. We believe that this result may inspire some future architecture design.

Figure 1: We extract the PONO statistics from VGG-19, ResNet-152, and Dense-161 at layers right before downsampling (max-pooling or strided convolution).

Figure 2: The generator of CycleGAN + PONO-MS. Pix2pix uses the same architecture. The operations in a block is applied from left to right sequentially. The **blue** lines show how the first two moments are passed. ConvTrans stands for transposed convolution. Each ResBlock has Conv3x3, BN, ReLU, Conv3x3, and BN.

Figure 3: Qualitative results of CycleGAN (with/without PONO-MS) with randomly sampled inputs.

| Real A | AtoB | AtoB with PONO-MS | Real B | Real A | AtoB | AtoB with PONO-MS | Real B |

Figure 4: Qualitative results of Pix2pix (with/without PONO-MS) with randomly sampled inputs.

## Footnotes

*: Equal contribution.

[2]https://github.com/bearpaw/pytorch-classification

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

| Inputs | | cat2dog (cat's content + dog's attributes) | | | | dog2cat: (dog's content + cat's attributes) | | | |
|---|---|---|---|---|---|---|---|---|---|
| Cat | Dog | DRIT | DRIT + PONO-MS | MUNIT | MUNIT' + PONO-MS | DRIT | DRIT + PONO-MS | MUNIT | MUNIT' + PONO-MS |

Figure 5: Qualitative results of DRIT and MUNIT (with/without PONO-MS) with randomly sampled inputs.

[9] Jun-Yan Zhu, Taesung Park, Phillip Isola, and Alexei A Efros. Unpaired image-to-image translation using cycle-consistent adversarial networks. In *Proceedings of the IEEE international conference on computer vision*, pages 2223–2232, 2017.

Figure 6: Training and validation curves of ResNet-18 and ResNet-18 + PONO on ImageNet.