[Reviews · NeurIPS 2019]

Reviewer 1



The paper is fairly well-written, the structure is clear. Experiments are strong and contain main image translation methods (pix2pix and cyclegan) as well as some new multimodal ones (DRIT, MUNIT). The authors compare with SPADE (basically spatially-aware BN that shares some ideas with PONO). I find this submission quite original and novel. I believe it will be useful for a variety of generative models. Implementation is straightforward (the code is given in supplementary) so I consider it reproducible.

Reviewer 2



The idea introduced in this paper is original. The writing is clearly written and easy to follow. The experiments are diverse with clear ablation study. The results are significantly improved for the proposed method.

Reviewer 3



[Final updated review] I updated my score from marginally below the acceptance threshold (5) to A good submission (7) due to three reasons. First, my concern for figure 5 is solved by the author response. Second, the author shows the other applications that PONO can help. Third, my concern for PONO itself is not solved. [First] I thought that figure 5 didn't cover the appropriate ablation study on PONO-MS, while the original methods in the paper (DRIT and MUNIT) are already equipped with adaptive instance normalization. [Second] Author response on reviewer 1 & 2 shows that PONO-MS can boost the performance of image dehazing, visual navigation, super-resolution and etc. Although I thought the contribution is incremental to the other normalization works, the fact that PONO-MS improves the performance of many application in a consistent manner insists that PONO-MS is robust, effective and universally applicable methods. [Third] Although I also agree that re-injecting the spatial information by MS can help the performance boost, my concern on the effectiveness of PONO itself (positional normalization itself) is not explained well. I conjecture that PONO itself have some regularization effects because the architectural setting that PONO removes amplitude information among filter responses enforces the structure information in the encoder is modeled as both mean/std statistics (for PONO-MS) and correlation among filter responses (for the next layer). ============================================================== The proposed positional normalization technique consists of two parts; positional normalization step (PONO) and moment shortcut (MS). The combination of two components is designed in a computationally-efficient manner compared to the SPADE module, which shows state-of-the-art performance. By analyzing the table 4, I think that the major advantage of the normalization comes from MS, not PONO. I think that PONO itself is an incremental work compared to the other normalization techniques such as BN, GN and IN because recently many normalizations works combine multiple normalizations into a single normalization to achieve a better estimation of statistics. Table 4 shows that PONO is worse than the other normalizations in several cases. So, I conclude that the contribution of PONO is rather weak.

[Author Response · NeurIPS 2019]

**Author Feedback of Positional Normalization**

We thank all reviewers for their insightful and constructive comments.

**Reviewer #1 and #2:**

Since the original paper submission, we have explored PONO in the context of other applications and model architectures. Although these results are still preliminary, they are consistently positive and highly encouraging. Precisely, we have also applied PONO-MS to:[1]

1) Image dehazing. We apply one PONO-MS to AOD-Net [4] using the offcial codebase and test on the same evaluation datasets provided in the paper. For TestSet A, the PSNR increases from 19.69 to 20.38 dB, the SSIM increases from 0.8478 to 0.8587. For TestSetB, the PSNR increases from 21.54 to 21.67 dB, the SSIM increases from 0.9272 to 0.9285.

2) Visual Navigation. Here we add one PONO-MS to the visual processing part of the official codebase on Habitat Platform [6]. The test (phase RGBD) accuracy increases from 0.79 to 0.81.

3) Super-resolution (suggested by Reviewer #1). We first build an autoencoder consisting of 20 convolutional layers and test on Set5. By adding PONO-MS, the PSNR increases from 34.75 to 35.60 dB (Scale=2), 32.04 to 32.94 dB (Scale=3) and 29.93 to 30.56 dB (Scale=4). Furthermore we take preliminary study and conduct experiments using VDSR [3], EDSR [5] and CARN [1] on Set5 and B100 datasets, the results show PONO-MS can improve the PSNR of the baseline models by about 0.03 to 0.08 dB (Scale=2, 3, 4), and the SSIM improves slightly. We will explore more models of these tasks.

4) Other tasks. We also take preliminary trials on 2D human pose estimation task based on two-stage approach of Mask R-CNN [2], the results show that PONO-MS is able to improve the average precision (AP) of both bounding-box person detection and person keypoint detection by around 1 point.

**Reviewer #3:**

We apologize in case Table 4 was confusing and will try to clarify it in the final version.

1) Unlike previous normalization methods such as BN and GN that emphasize on accelerating and stabilizing the training of networks, PONO is used to split off part of the spatial information and re-inject it later. Therefore, PONO-MS is complimentary to these normalization methods.

2) Table 4 aims to show PONO-MS could be compatible with (or be applied together with) other normalization methods. <BN + PONO-MS> is simply applying <PONO-MS> to the baseline model and keep the original BN modules which have a different purpose: to stabilize and speed up the training. We also show the models where BN is replaced by LN/IN/GN as well as these models with PONO-MS. We show that PONO-MS can be applied jointly with each one of them and improve performance. The last row shows PONO-MS can work independently when we remove the original BN in the model.

3) Instance Normalization (IN) and PONO have their own usages and advantages. Recent GAN models, such as MUNIT and StyleGAN, use Adaptive IN (AdaIN) to control the style of the generated images. In contrast, PONO-MS is focusing on controlling the structural information — for example, when we want to translate a dog to a cat image, using a dog image A and a reference cat image B. MUNIT would extract the style information from the reference B and pass it to AdaIN, but PONO is applied to extract the structural information from the source A instead.

4) Thanks for your suggestion of using <BN-MS>, we will take this into consideration and explore it. While BN computes the activations over the positions and doesn't align with our story of isolating spatial information, we believe that it may be worth exploring for other applications.

## Footnotes

[1]All experiments are on the basis of the same settings (including datasets) from the official papers or codebase. For task 1) and 3), we utilize the two most popular evaluation metrics PSNR and SSIM. For these scores, the higher the better.

# References

[1] Namhyuk Ahn, Byungkon Kang, and Kyung-Ah Sohn. Fast, accurate, and lightweight super-resolution with cascading residual network. In *Proceedings of the European Conference on Computer Vision (ECCV)*, pages 252–268, 2018.

[2] Kaiming He, Georgia Gkioxari, Piotr Dollár, and Ross Girshick. Mask r-cnn. In *Proceedings of the IEEE international conference on computer vision*, pages 2961–2969, 2017.

[3] Jiwon Kim, Jung Kwon Lee, and Kyoung Mu Lee. Accurate image super-resolution using very deep convolutional networks. In *Proceedings of the IEEE conference on computer vision and pattern recognition*, pages 1646–1654, 2016.

[4] Boyi Li, Xiulian Peng, Zhangyang Wang, Jizheng Xu, and Dan Feng. Aod-net: All-in-one dehazing network. In *Proceedings of the IEEE International Conference on Computer Vision*, pages 4770–4778, 2017.

[5] Bee Lim, Sanghyun Son, Heewon Kim, Seungjun Nah, and Kyoung Mu Lee. Enhanced deep residual networks for single image super-resolution. In *Proceedings of the IEEE conference on computer vision and pattern recognition workshops*, pages 136–144, 2017.

[6] Manolis Savva, Abhishek Kadian, Oleksandr Maksymets, Yili Zhao, Erik Wijmans, Bhavana Jain, Julian Straub, Jia Liu, Vladlen Koltun, Jitendra Malik, et al. Habitat: A platform for embodied ai research. *arXiv preprint arXiv:1904.01201*, 2019.



[Meta-Review · NeurIPS 2019]

A new type of normalization for convnets across channels. Simple idea but strong experiments. Would be of immediate interest for many attendees / readers.